# Improved Bidirectional RRT* Algorithm for Robot Path Planning

**DOI:** 10.3390/s23021041

**Published:** 2023-01-16

**Authors:** Peng Xin, Xiaomin Wang, Xiaoli Liu, Yanhui Wang, Zhibo Zhai, Xiqing Ma

**Affiliations:** Key Laboratory of Intelligent Industrial Equipment Technology of Hebei Province, School of Mechanical and Equipment Engineering, Hebei University of Engineering, Handan 056038, China

**Keywords:** artificial potential field method, improved bidirectional RRT* algorithm, dynamic window method, fusion algorithm, temporary obstacles avoidance

## Abstract

In order to address the shortcomings of the traditional bidirectional RRT* algorithm, such as its high degree of randomness, low search efficiency, and the many inflection points in the planned path, we institute improvements in the following directions. Firstly, to address the problem of the high degree of randomness in the process of random tree expansion, the expansion direction of the random tree growing at the starting point is constrained by the improved artificial potential field method; thus, the random tree grows towards the target point. Secondly, the random tree sampling point grown at the target point is biased to the random number sampling point grown at the starting point. Finally, the path planned by the improved bidirectional RRT* algorithm is optimized by extracting key points. Simulation experiments show that compared with the traditional A*, the traditional RRT, and the traditional bidirectional RRT*, the improved bidirectional RRT* algorithm has a shorter path length, higher path-planning efficiency, and fewer inflection points. The optimized path is segmented using the dynamic window method according to the key points. The path planned by the fusion algorithm in a complex environment is smoother and allows for excellent avoidance of temporary obstacles.

## 1. Introduction

Mobile robots have a wide range of applications in industry, agriculture, and daily life. To enable robots to quickly reach their target point and complete a designated task in a complex environment, an excellent path-planning algorithm should be employed to plan an effective path. Therefore, a path-planning algorithm is a core facet of the development of mobile robots [1,2]. Common path-planning algorithms include the artificial potential field method [3], ant colony algorithms [4], the A* algorithm [5], the dynamic window method [6], etc.

The RRT algorithm is a classical path-planning algorithm [7]. The main idea of the algorithm is to find an effective path by growing a random tree continuously until the target point is also on the random tree. The defects of the RRT algorithm are obvious. The RRT algorithm randomly obtains sampling points when looking for a path, which makes its search efficiency low, and there are many redundant nodes and redundant sections in the planned path. RRT* [8] and bidirectional RRT* are commonly used to improve the RRT algorithm. At present, scholars have carried out in-depth research on the RRT algorithm. Mashayekh et al. [9] proposed Informed RRT*-Connect, which improved the sampling of RRT*-Connect and provided better solutions. In [10], the author proposes EB-RRT, which facilitates a robot’s movement in dynamic environments using heuristics to plan a global path and the EB method to optimize the heuristic path. In [11], the authors propose a new rewiring method based on triangulation to improve the RRT algorithm. The planning time is closer to the optimum than the traditional RRT algorithm. In [12], a regression mechanism is added to RRT to prevent over-searching. The algorithm uses adaptive expansion to avoid duplicate searches and improve search efficiency by refining the boundary nodes. In [13], the RRT*-AB algorithm is proposed. This algorithm can quickly determine the target area, adjust bounds, and improve computational efficiency. Qi [14] proposes the MOD-RRT* algorithm for unknown dynamic environments. The path is first planned using the improved RRT* algorithm; then, the initial path is optimized. Wang [15] used a convolutional neural network to optimize the RRT* algorithm. The prediction of optimal paths by CNN models guides the RRT* algorithm’s sampling, thus improving path-planning efficiency. The authors of [16] propose an LM-RRT algorithm that uses reinforcement learning to solve path planning in narrow environments. Qie [17] proposed the (TF-RRT*) method. This algorithm only allows the sampling points to be biased towards the target point, and the random numbers grow more biased towards the target point, which effectively improves path-planning efficiency.

The authors of [18] improved search efficiency by adjusting the direction of growth tree expansion with situational data. These algorithms generally require adjustments and place restrictions on the search direction of the random tree and the selection of sampling points.

In this paper, we propose a new algorithm combining the bidirectional RRT* algorithm and the artificial potential field method, which can accelerate the combination of two trees during the sampling process and thus effectively improve path-planning efficiency. Combining the new algorithm with the dynamic window method in the global path enables a robot to reach a target point safely and avoid temporary obstacles.

## 2. Relate Work

### 2.1. Principle of RRT*

The main aim of RRT is to grow a random tree at a starting point and then grow a random point in blank space. The process transpires as follows: find the point in the random tree that is nearest to the random point, and then extend one step from the nearest point to the random point. When the extended step does not lead to successful navigation through the obstacles, new nodes are successively added to the random tree until the target point is also in the random tree and the path is found; consequently, the path-planning process is completed. Although the RRT algorithm can basically plan an effective path, its search efficiency will be greatly reduced due to the approximate uniform expansion of the random tree in the surrounding space.

The RRT* algorithm improves the process of re-selecting the parent node on the basis of the RRT algorithm. By constantly adjusting the parent node, the path is gradually optimized. This path optimization procedure will increase the path-planning time of the RRT* algorithm. The specific process is shown in Figure 1.

Figure 1a shows the path planned by the RRT algorithm and Figure 1b shows the path planned by the RRT* algorithm. In Figure 1, node ③ is X_new and the path planned by the RRT algorithm is S->①->②->③. With node ③ as the center of the circle, a circle is drawn with a certain distance as the radius; nodes ②, ④ and ⑤ are within the circle. When comparing path S->①->②->③ and path S->④->③, path S->④->③ has the shortest path length. At this stage, the path is replanned, with node ④ as the parent of node ③; the replanned path is shown in Figure 1b.

### 2.2. Principle of Bidirectional RRT*

When using a single random tree to search the whole space, the search efficiency is low. Using the RRT* algorithm to expand the random tree at both the start and target side can effectively improve search efficiency. When two trees are connected or when two trees are within a certain distance of each other, the search is completed, as shown in Figure 2.

In Figure 2, the red line indicates the connection of the two trees growing from the start point and target point, indicating that the global valid path has been found. The global path is as follows: S->④->③->ⓓ->ⓐ->T.

## 3. Improved Bidirectional RRT* Algorithm

Although the bidirectional RRT* algorithm has been greatly improved compared to the traditional RRT algorithm, there are still some problems during the expansion of the two trees, such as the excessively high degree of randomness of the sampling points, the low path-planning efficiency, and the many inflection points and redundant sections in the planned path. Therefore, the following improvements are proposed for the bidirectional RRT* algorithm.

### 3.1. Adding Artificial Potential Field Ideas

Although the traditional bi-directional RRT* algorithm grows random trees at both the start and target sides, the efficiency of its path planning is still low due to the random sampling points. Thus, the artificial potential field bias is increased to the first tree expansion, making the sampling more directional and improving search efficiency.

The artificial potential field gravitational function is as follows:(1)$$U_{att}=\frac{1}{2}\xi \rho^{2}(P,P_{goal})$$

In Equation (1), ξ is the gravitational coefficient and $\rho (P,P_{goal})$ is the distance between the current point and the target point.

The repulsion function is as follows:(2)$$U_{rep}=\left\{\begin{array}{ll} \frac{1}{2}\mu {\left(\frac{1}{\rho (P,P_{obs})}-\frac{1}{\rho_{0}}\right)}^{2}, & \rho (P,P_{obs})\leq \rho_{0}\\ 0 & ,\rho (P,P_{obs})>\rho_{0} \end{array}\right.$$

In Equation (2), μ is the repulsion coefficient and $\rho (P,P_{obs})$ is the distance between the current point and obstacle. $\rho_{0}$ is the influence range of the obstacle. Beyond this range, the obstacle has no repulsive effect on the current node.

Let the combined potential fields be
(3)$$U=U_{att}+U_{rep}$$

When the traditional artificial potential field method is used to calculate the joint potential field of the nodes around the current position, the search range is too small, which is not conducive to the determination of the surrounding nodes. Therefore, an extended search range is implemented. If the search range is too large in a complex environment, it can reduce the efficiency of path planning; therefore, the search range was changed to 5 ∗ 5, as shown in Figure 3.

The 5 ∗ 5 search range artificial potential field method was added to the selection of sampling points, as shown in Figure 4.

In Figure 4, P1_init is the current node, P1_APF is the artificial potential field method’s bias of the current node, P1_rand is the randomly sampled node, P1_step is the node extended by one step, and P1_real is the actual extended node. The artificial potential field method is first used to improve the sampling points; then, the nodes in the path are re-selected as parents to further optimize the path.

### 3.2. Adjusting the Sampling Direction of a Random Tree Growing at a Target Point

The random tree growing at the target side is biased towards the sampling point of the starting tree when it is expanded, as shown in Figure 5.

In Figure 5, P1_real is the actual sampling point of the random tree grown at the starting point, P2_init is the closest point to P1_real, which is growing at the end point. P2_step is an extension of one step towards P2_init to P1_real. The extension of the random tree growing at the end point is deflected to the actual sampling point of the random tree growing at the starting point, thereby accelerating the connection of the two trees and improving path-planning efficiency.

The improved bidirectional RRT* algorithm was compared with the traditional bidirectional RRT* algorithm in an obstacle-free and an obstacle-strewn environment to verify the effectiveness of the improvements. A comparison of the two algorithms in the obstacle-free environment is shown in Figure 6.

Figure 6a shows the path planned by the bidirectional RRT* in the obstacle-free environment and Figure 6b shows the path planned by the improved bidirectional RRT*. By comparing Figure 6a,b, it is evident that the improved bidirectional RRT* planning results in significantly shorter paths, smoother paths, and fewer sampling points in the obstacle-free environment.

When there are obstacles in the test environment, if the actual sampling point is on the obstacle, the point is dropped, and the search is repeated. Figure 7 shows a comparison of the two algorithms when an obstacle is present in the environment.

Figure 7a shows the path planned by the bidirectional RRT* in an obstacle-strewn environment and Figure 7b shows the path planned by the improved bidirectional RRT*. By comparing Figure 7a,b, it is evident that the improved bi-directional RRT* algorithm plans shorter paths with fewer sampling points in the presence of obstacles, which is a significant improvement compared to the traditional bidirectional RRT* algorithm.

Comparing the two algorithms in the presence and absence of obstacles, the path length as well as the direction of the random tree’s expansion and the number of sampling points show that the improved bidirectional RRT* algorithm plans shorter and more biased paths when expanded.

### 3.3. Path Optimization

There are still redundant nodes and redundant sections in the path planned by the improved bidirectional RRT*. A diagram of the redundant nodes is shown in Figure 8.

In Figure 8, the line between points A and C does not pass through the obstacle, in which case point B is the redundant node.

At this time, the path planned by the improved bidirectional RRT* algorithm is optimized to extract key nodes and eliminate redundant nodes. The specific steps of the process are as follows:Put all the nodes into the set $\left\{P_{1},P_{2},P_{3}\ldots P_{n}\right\}$
in order.Connect the nodes in the set one by one from the starting node $P_{t}$ until the connection between the node with $P_{t+1}$ passes the obstacle and $P_{t}$ is the key point in the set. At this point, starting from $P_{t}$, connect the remaining nodes in turn until all the key points are found.Connect the key points and target points in sequence from the starting point to plan the new path, as shown in Figure 8.

Figure 9b shows the key points on the path planned by the improved bidirectional RRT* in an obstacle-strewn environment and Figure 9c shows the re-routed path according to the key nodes.

The formula for calculating the length of the path is as follows:(4)$$L=\sqrt[{2}]{{\left|x_{1}-x_{2}\right|}^{2}+{\left|y_{2}-y_{2}\right|}^{2}}$$

Keep accumulating the distance between each of the two nodes until finally obtaining the length of the whole path.

As can be seen in Table 1, the optimization procedure has resulted in a 7.47% reduction in path length. There is a 77.8% reduction in the number of inflection points in the path. The optimized path is smoother, with fewer inflection points and a shorter path length.

## 4. The Incorporation of the Dynamic Window Method

The dynamic window method simulates the path of a robot at all speeds in the velocity space [19,20]; scores the robot’s orientation, movement speed, and the distance to obstacles over a certain period of time; and aggregates the scores according to a certain ratio to obtain the path with the highest total score. Naturally, the path with the highest score is the best path.

### 4.1. Robot Kinematic Models

Assuming that, over a short period, the robot moves in a straight line with an angular velocity of *ω* and a linear velocity of *v* during movement, the motion of the robot in Δ*t* can be described as follows:(5)$$\left\{\begin{array}{l} x_{t+△t}=x_{t}+\nu △t\cos(\theta_{t})\\ y_{t+△t}=y_{t}+\nu △t\sin(\theta_{t})\\ \theta_{t+△t}=\theta_{t}+\omega △t \end{array}\right.$$

Using Equation (5) and the velocity information collected, the trajectory of the robot can be simulated over the next △t time.

### 4.2. Velocity Sampling

There are a large number of angular and linear velocities in velocity space. At this point, the velocity should be constrained, accounting for the robot’s motor situation and the current environment, as in Equation (6).
(6)$$V_{m}=\left\{\left(v,\omega \right)|v\in \right.[v_{min},\;v_{max}],\omega \in \left[\omega_{min},\;\omega_{max}\right]\}$$

Motor torque and other factors will limit the movement speed of the robot. In a real environment, the moving speed of the robot is as follows:(7)$$V_{d}=\left\{\left(v,\omega \right)\right.|v\in \left[v_{c}-{\dot{v}}_{b}\Delta t,v_{c}+\overset{}{{\dot{v}}_{a}\Delta t}\right],\omega \in \left[\omega_{c}-{\dot{\omega }}_{b}\Delta t,\omega_{c}+\overset{}{{\dot{\omega }}_{a}\Delta t}\right]\left.\right\}$$

In Equation (7): the current velocity is $v_{c}$, $\omega_{c}$; the maximum acceleration is ${\dot{v}}_{a}$, ${\overset{{}_{.}}{\omega }}_{a}$; and the maximum deceleration is ${\dot{v}}_{b}$, ${\dot{\omega }}_{b}$.

To enable the robot to stop in time before it hits an obstacle, set an upper speed limit for the robot at:(8)$$v_{a}=\left\{(v,\omega )\left|v\leq {\left(2dist(v,\omega ){\dot{v}}_{b}\right)}^{\frac{1}{2}},\right.\omega \leq {\left(2dist(v,\omega ){\dot{\omega }}_{b}\right)}^{\frac{1}{2}}\right\}$$

In Equation (8), dist(v,ω) is the shortest distance between the simulated path and the obstacles. There are multiple sets of feasible trajectories in the sampled velocities, and it is necessary to use an evaluation function to select the optimal path.
(9)G(v,ω)=σ(αhead(v,ω)+βdist(v,ω)+γvel(v,ω))

In Equation (9), head(v,ω) is used to evaluate the deviation of the target direction and the simulated trajectory direction at the current velocity. vel(v,ω) is used to evaluate the linear velocity of the robot. σ is the smoothing function and α, β, and γ are the coefficients.

The traditional dynamic window method can fall into a local optimum and thus fail to find a valid path in operation, as shown in Figure 10.

As can be distinct seen in Figure 10a,b, the dynamic window method easily falls into a local optimum when there are corners in the ground, thus failing to find a globally valid path, so, in practice, the dynamic window method is used as a local algorithm. The improved bi-directional RRT* algorithm is used as a global algorithm combined with the local algorithm.

## 5. Fusion Algorithm

The traditional dynamic window method is prone to fall into local optimality when planning paths and requires a global path-planning algorithm to guide it. Global planning is performed using the improved RRT algorithm, which segments the global path according to key points and applies the dynamic window method to each segment of the path [21,22]. In the global algorithm, the improved RRT plans the global optimal path; in the local algorithm, the improved RRT algorithm guides the dynamic window method to plan the path, thereby preventing the dynamic window method from falling into a local optimum and failing to find the path. The schematic diagram of the fusion algorithm is shown in Figure 11.

In Figure 11, P1 and P2 are two adjacent key nodes, and the paths between the two critical nodes are planned using the dynamic window method. In order to understand the fusion algorithm flow more clearly, a flow chart of the fusion algorithm has been constructed, as shown in Figure 12.

## 6. Simulation Verification

To test the efficiency of the improved bidirectional RRT* algorithm and the effectiveness of the fusion algorithm, a matrix was used in MATLAB 2017 to establish a grid graph, with black squares representing obstacles and white squares representing clear space. The starting node is at (4,3); the target point is at (29,24). Two groups of experiments were used for detection and comparison.

Experiment 1: A comparison the path-planning performance of the, the traditional RRT, the traditional A*, the traditional bidirectional RRT*, the improved bidirectional RRT* algorithm, and the fusion algorithm is shown in Figure 13a–e.

Several algorithm extensions and sampling steps can be clearly seen in Figure 13. To better verify the effectiveness of the algorithm, a quantitative comparison was performed using the table. The five algorithms’ plan times, path lengths, and number of inflection points are shown in Table 2.

In Table 2, these five algorithms are compared in terms of path length; it is evident that the fusion algorithm plans the shortest path length. The fusion algorithm reduces the path length by 0.59% compared to that of the improved bidirectional RRT* algorithm. In terms of path-planning time, the improved bidirectional RRT* algorithm uses the shortest time to plan an efficient path. In terms of path inflection points, except for the fusion algorithm, the improved bidirectional RRT* algorithm plans the path with the least number of inflection points and plans the smoothest path.

Experiment 2: Temporary obstacles were added to the global path planned by the improved bidirectional RRT* algorithm, as shown by the red squares in Figure 14, to test whether the fusion algorithm enables the robot to avoid temporary obstacles.

By adding temporary obstacles to the global trajectory, the robot moves toward the target point as well as the obstacle avoidance state, as shown in Figure 15.

As shown in Figure 15, the path planned by the fusion algorithm effectively bypasses the two temporary obstacles and the path does not intersect with the obstacles in the figure, allowing the robot to reach the target point safely. This proves that the fusion algorithm is globally effective and that the planned path is able to avoid temporary obstacles.

Figure 16a shows the robot’s linear movement speed and Figure 16b shows the angular velocity of the robot’s movement. The robot reaches the target point safely, as evidenced by the velocity variation. Thus, the fusion algorithm has proven to be globally effective.

## 7. Conclusions

This paper offers two contributions. (1) The concept of the artificial potential field method was incorporated into the bidirectional RRT* algorithm to adjust the sampling points of random numbers generated at the starting point and the target point. Through simulation and comparison with the traditional A*, the traditional RRT, and the traditional RRT* algorithms, it has been verified that the improved bidirectional RRT* algorithm offers shorter path planning and higher path-planning efficiency. (2) The improved bi-directional RRT* algorithm was combined with the dynamic window method. It has been proven that the fusion algorithm can plan an effective path and avoid temporary obstacles. The algorithm also has some disadvantages, as its bias is more pronounced; therefore, it is not suitable for planning efficient paths in very complex environments.

The algorithm can be used in ROS mobile robots using the bidirectional RRT* algorithm as the global path algorithm. The path-planning algorithm will be explored in depth in future research for application in real robots.

## Figures and Tables

**Figure 1 sensors-23-01041-f001:**
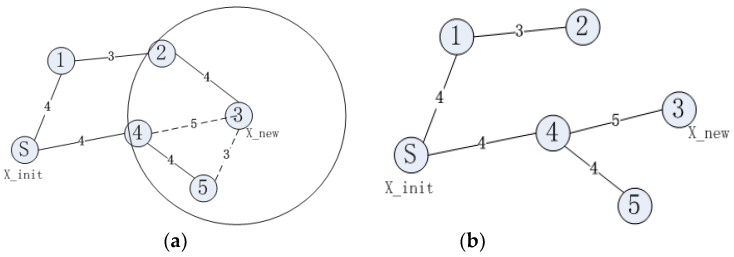
Principle of RRT* extension (**a**) Path planned by traditional RRT algorithm, (**b**) Optimized path.

**Figure 2 sensors-23-01041-f002:**
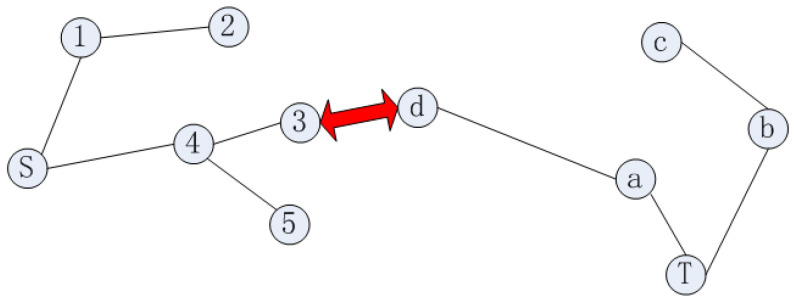
Principle of bidirectional RRT* extension.

**Figure 3 sensors-23-01041-f003:**
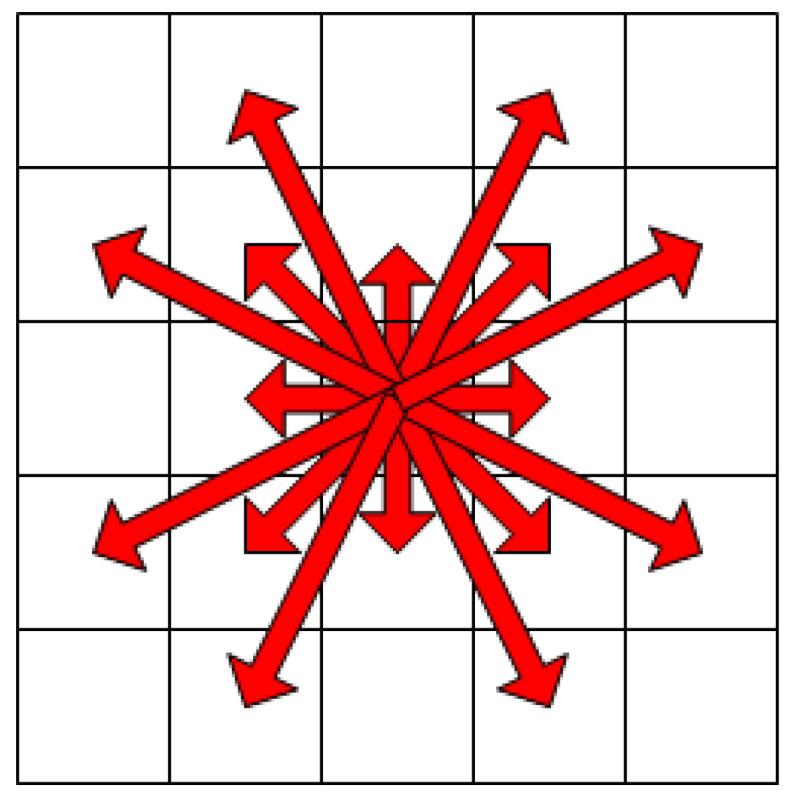
5 ∗ 5 search range.

**Figure 4 sensors-23-01041-f004:**
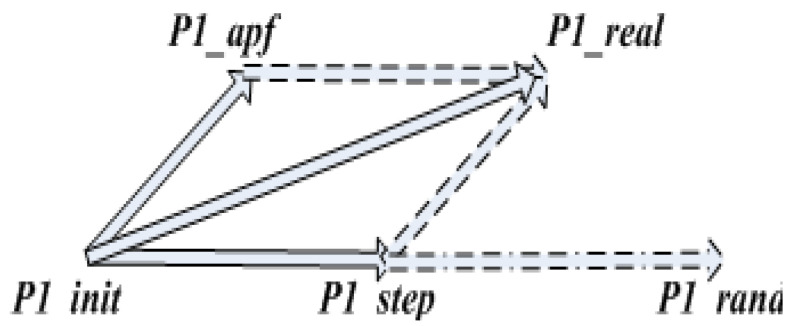
Sampling with the addition of artificial potential field concepts.

**Figure 5 sensors-23-01041-f005:**
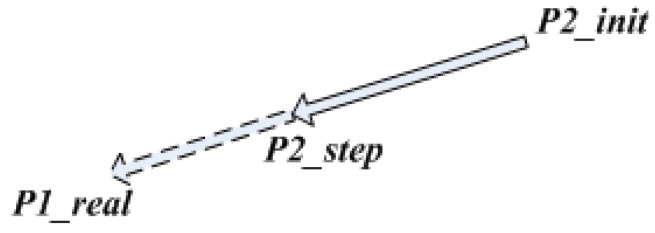
The sampling direction of a random tree growing at target point.

**Figure 6 sensors-23-01041-f006:**
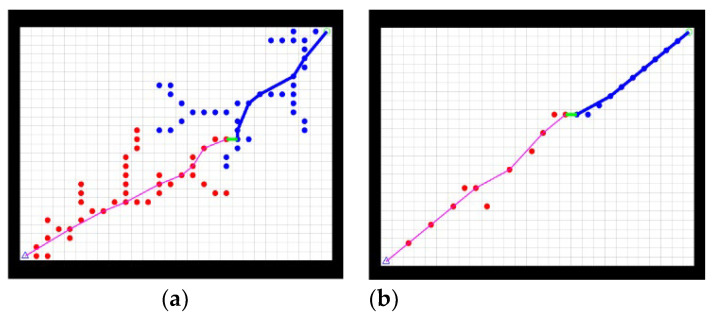
Comparison of two algorithms in obstacle-free environment. (**a**) Traditional bidirectional RRT* algorithm, (**b**) Improved bidirectional RRT* algorithm.

**Figure 7 sensors-23-01041-f007:**
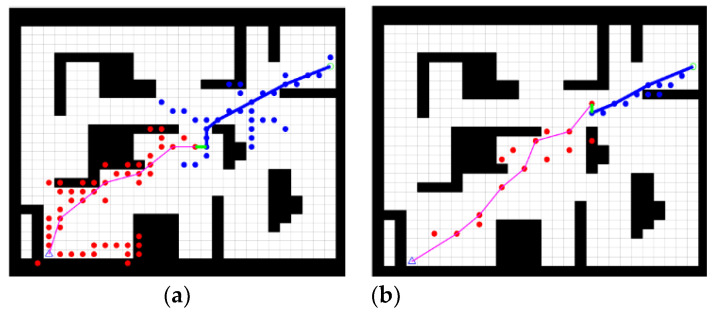
Comparison of two algorithms in obstacle-strewn environment. (**a**) Traditional bidirectional RRT* algorithm, (**b**) Improved bidirectional RRT* algorithm.

**Figure 8 sensors-23-01041-f008:**
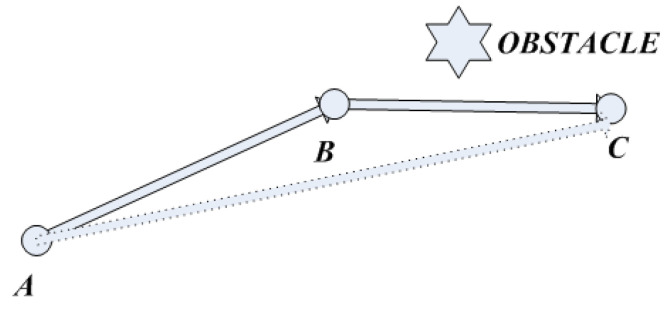
Diagram of the redundant nodes.

**Figure 9 sensors-23-01041-f009:**
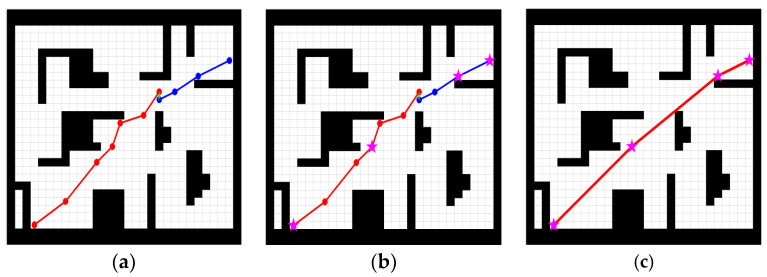
Optimization steps. (**a**) Original path, (**b**) Extraction of key nodes, (**c**) Final Path.

**Figure 10 sensors-23-01041-f010:**
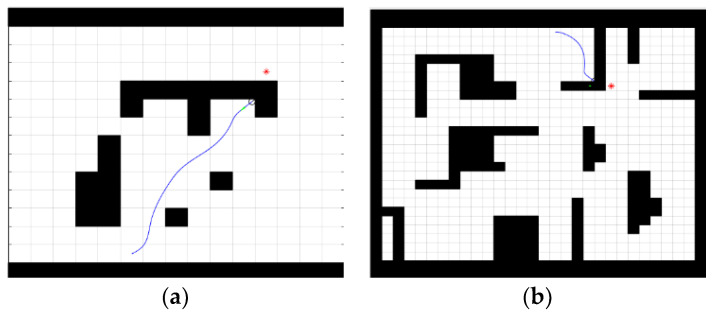
Traditional dynamic window method-based planning path. (**a**) 15 ∗ 15 map, (**b**) 30 ∗ 30 map.

**Figure 11 sensors-23-01041-f011:**
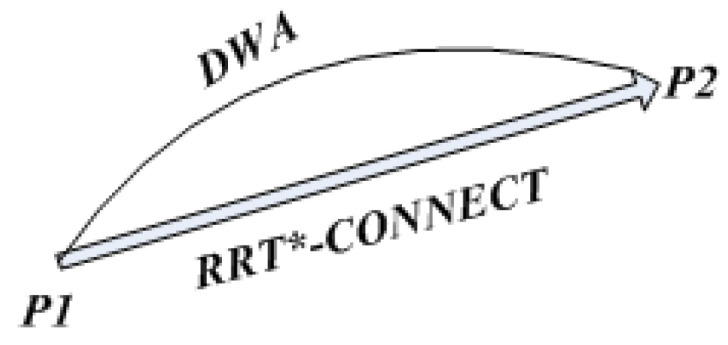
Diagram of fusion algorithm.

**Figure 12 sensors-23-01041-f012:**
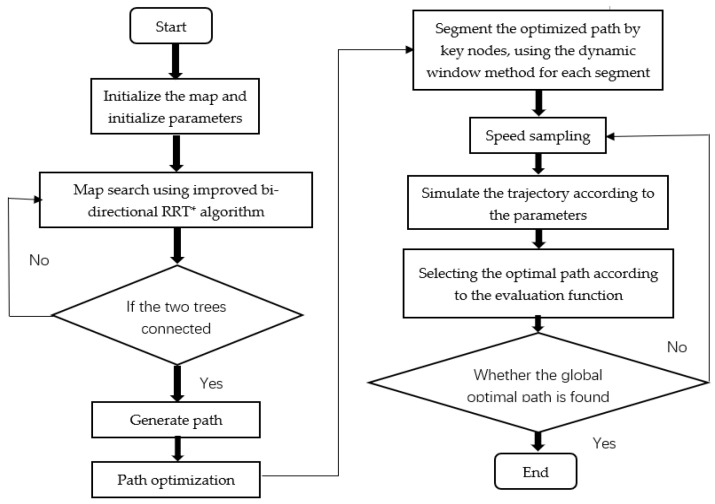
Flow chart of fusion algorithm.

**Figure 13 sensors-23-01041-f013:**
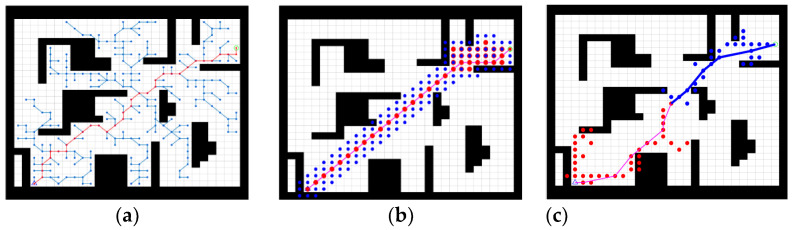
Five algorithms for planning the path. (**a**) Traditional RRT, (**b**) Traditional A*, (**c**) Traditional bidirectional RRT*, (**d**) Improved bidirectional RRT*, (**e**) Fusion algorithm.

**Figure 14 sensors-23-01041-f014:**
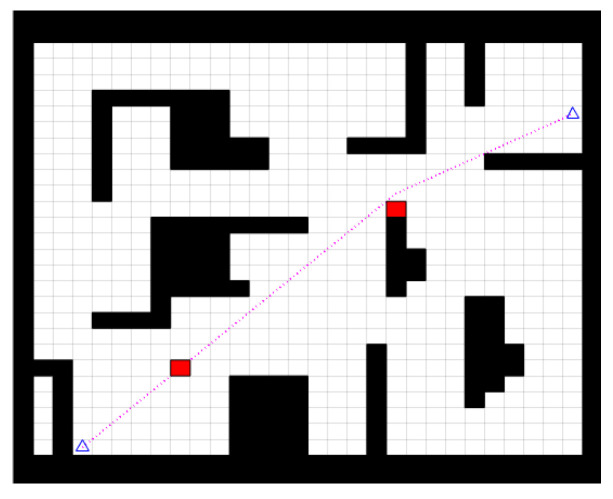
The addition of temporary obstacles to the global path.

**Figure 15 sensors-23-01041-f015:**
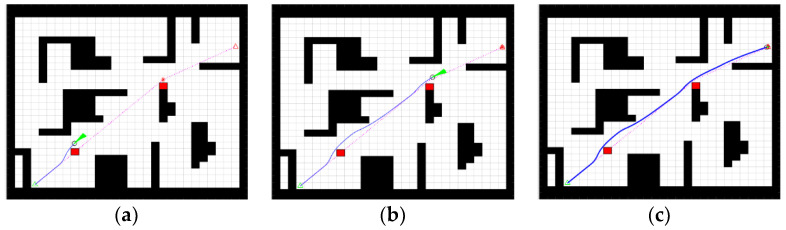
Overall trajectory. (**a**) First avoidance, (**b**) Second avoidance, (**c**) Reaching the target.

**Figure 16 sensors-23-01041-f016:**
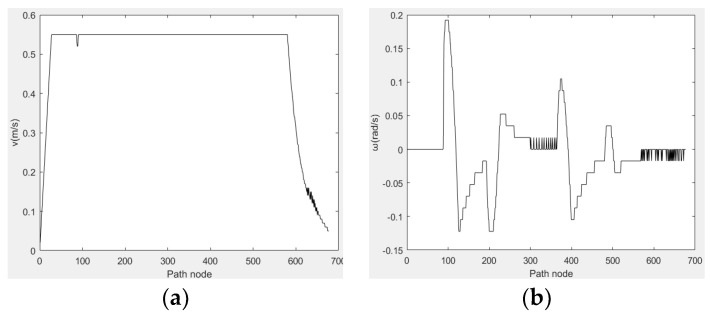
Speed variation in robot’s movement. (**a**) Line Speed, (**b**) Angular velocity.

**Table 1 sensors-23-01041-t001:** Comparison of improved RRT algorithm and optimized path.

Path-Planning Algorithms	Path Length	Number of Path Inflection Points
Improved bidirectional RRT*	35.4754	9
After optimizing the path	32.8269	2

**Table 2 sensors-23-01041-t002:** Comparison of five algorithms.

Path-Planning Algorithms	Path Length	Planning Time(s)	Number of Path Inflection Points
Traditional RRT	39.2132	1.3745	25
Traditional A*	33.6985	0.4971	2
Traditional bidirectional RRT*	35.0956	0.1178	10
Improved bidirectional RRT*	32.9230	0.0513	1
Fusion algorithm	32.7300	377.0923	none

## Data Availability

The data presented in this study are available on request from the corresponding author.

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
