# Peer review of "Improved Bidirectional RRT* Algorithm for Robot Path Planning"

_sensors, 2023, doi:10.3390/s23021041_

Round 1

Reviewer 1 Report

In summary, the contribution of this article is relevant because the proposed hybrid algorithm seems to handle every drawback of traditional path-planning algorithms (being stuck in local minima, planning time in the order of seconds and the existence of redundancy and inflection points in the search graph). Thus, the contribution is successful. It is worth mentioning that the results shown by the Figure 15 are elegantly exposed and it is really interesting the achieved reactive control capabilities. Also, the results are good and it is clear that the algorithm provides a smoother and optimal trajectory. Finally, it is really valuable that the trajectory is adaptable (it can adapt to the appearance of temporary obstacles) and the planning time is relatively short.

Regarding the form of the article, I have the following comments: In section 3.2, line 165, I assume that the figures 7c) and 7d) were mistakenly referred, when 7a) and 7b) should be referred instead. In section 3.3. (lines 173 and 174) I felt that some illustrations of the mentioned redundant nodes and sections could be provided. I also had to re-read several times the steps for the optimization of improved bidirectional RRT* in the section 3.3. I suppose the explanation could be more clearer or pseudocode could be provided instead. On lines 187 and 188 I assume that the figures were mistakenly referred. From table 1, I think it should be exposed h ow the Path length metric is computed by an inline equation. In Figure 10, there is a mistake: the "Generate path" step is represented twice. In lines 261-263 I assume there is a mistake because you mention 6 algorithms, but the figure only represents 5 algorithms. In lines 278-279 it could have been referred the percentage by how much the fusion algorithm improved the path length over the improved bidirectional RRT* algorithm. In figure 14, I would caption explicitly each sub figure, like: "a ) First avoidance, b) Second avoidance, c) Reaching the target". Furthermore, I am confused about the objectives of the work. It seems that your work has two contributions: (i) Improved bidirectional RRT*, by combining the traditional bidirectional RRT* with artificial potential field principles; (ii) A fusion algorithm that combines the improved bidirectional RRT* with the dynamic window method. Are both contributions being considered separately or should they be merged into a unique contribution? To conclude, I pose a final question: is this algorithm feasible to be applied online since the planning time is sufficiently short? This should be mentioned in the manuscript.

Author Response

Your suggestions have greatly improved my article. I didn't notice a lot of things. I am really glad that you are my reviewer. The revised parts are listed in detail in the attachment.

Reviewer 2 Report

There are many misunderstandings in this work, both on the substantive and editorial side. Below are comments for the authors of this work.

1.     I propose to expand the literature review with the latest references – 2022.

2.     The introduction is weak, I suggest broadening the literature review.

3.     On line 194 it is unclear, please correct. 4. Add dynamic window method(添加参考文献)

4.     Most equations need re-editing.

5.     Figure 8 is low quality and there is a bug, please modify it.

6.     The analysis of the results is not well presented, for example in Table 2 which does not properly describe the results. Please refer to this comment.

7.     I propose to extend the discussion of the results in order to confirm the correctness of the proposed solution and to point out the disadvantages of this method.

8.     I suggest expanding the references with articles from 2022.

Author Response

Your suggestions have greatly improved my article. I find that I do have many shortcomings. I have made detailed modifications to the paper, and I am glad that you are my reviewer. The detailed modifications are in the attachment。

Round 2

Reviewer 2 Report

Overall, the authors of the article responded to the comments, but could have better provided a literature review in the introduction.

In table 2, the scheduling time for the Fusion algorithm is 337.0923 s, if so, this algorithm is not suitable. let me explain.

Author Response

I apologize for not explaining in the article why the fusion algorithm takes so long, and thank you for asking this question.My answers are in the attachment.
